# Lactic Acid Bacteria (LAB): Autochthonous and Probiotic Microbes for Meat Preservation and Fortification

**DOI:** 10.3390/foods11182792

**Published:** 2022-09-10

**Authors:** Dibyajit Lahiri, Moupriya Nag, Tanmay Sarkar, Rina Rani Ray, Mohammad Ali Shariati, Maksim Rebezov, Sneh Punia Bangar, José M. Lorenzo, Rubén Domínguez

**Affiliations:** 1Department of Biotechnology, University of Engineering & Management, Kolkata 700160, India,; 2Department of Food Processing Technology, Malda Polytechnic, West Bengal State Council of Technical Education, Government of West Bengal, Malda 732102, India; 3Department of Biotechnology, Maulana Abul Kalam Azad University of Technology, Haringhata 741249, India; 4Semey Branch of the Institute, Kazakh Research Institute of Processing and Food Industry, 238«G» Gagarin Ave., Almaty 050060, Kazakhstan; 5Department of Scientific Research, Russian State Agrarian University-Moscow Timiryazev Agricultural Academy, 127550 Moscow, Russia; 6Department of Scientific Research, V. M. Gorbatov Federal Research Center for Food Systems, 26 Talalikhina St., 109316 Moscow, Russia; 7Biophotonics Center, Prokhorov General Physics Institute of the Russian Academy of Science, 38 Vavilov St., 119991 Moscow, Russia; 8Department of Food, Nutrition and Packaging Sciences, Clemson University, Clemson, SC 29634, USA; 9Centro Tecnológico de la Carne de Galicia, Avd. Galicia Nº4, Parque Tecnológico de Galicia, San Cibrao das Viñas, 32900 Ourense, Spain; 10Área de Tecnoloxía dos Alimentos, Facultade de Ciencias, Universidade de Vigo, 32004 Ourense, Spain

**Keywords:** food safety, food fortification, food microbiology, shelf-life, antimicrobial metabolites

## Abstract

The enhanced concern of the consumers regarding the safety, quality of the food products, and avoidance of the use of chemical food preservatives has resulted in a breakthrough in biopreservation. This has resulted in the use of beneficial microbial species, including bacteria and their secondary metabolites, to enhance the shelf-life and quality of the food products. Meat preservation and fortification are among the biggest concerns, as they are relevant to the majority of food products. The chemical preservatives conventionally used in preserving meat and meat products possess several detrimental effects on the consumers. Thus, alternative strategies are needed to combat strategically in facilitating the shelf-life and quality. Lactic acid bacteria (LAB) are considered the safest organism and have a profound role in food and food-processing industries. The biofilm developed by the bacteria prevents the growth of various undesirable microorganisms on meat and meat products. Various studies depicted that LAB produces various antimicrobial metabolites that can act effectively on the food-degrading pathogens, rendering it safe and enhancing shelf-life. This review, thus, deals with the use of LAB as biopreservatives for enhancing the shelf-life of meat and meat products and helping its fortification.

## 1. Introduction

The growing refusal of the consumers toward using food with chemical preservatives and additives has forced the food industries to use biopreservatives for food safety. The additives are used to maintain the quality and freshness of food, fortify or add nutritional value, and enhance the palatability to improve the taste or appearance of food. These days, beneficial bacteria and their metabolites, as potential natural preservatives for shelf life extension, are the most preferred option [1]. The use of meat and various meat products has always been an important part of the diet for humans, as it contains various essential nutrients that support health and growth. However, in most cases, the meat and meat products are susceptible to contamination by microorganisms, resulting in enhanced health risks for the customers and economic loss for the industry [2]. Among processed food, meat and meat products represents a serious challenge for the food industry, as the possible microbial contamination of fresh meat and meat products by various harmful bacteria, such as *Listeria monocytogenes*, cannot be handled by physical ways only, such as lowering the pH, freezing, and salting. Hence, such problems are produced by common food degrading microorganisms. One of the most used forms to address such problems is the use of lactic acid bacteria (LAB) and intelligent use of their antimicrobial properties, including production of bacteriocins and production of primary metabolite lactic acid, which in turn can decrease the pH, inhibiting the growth of a wide variety of food spoilage organisms [3].

LAB can grow in various habitats, including fermented meat, vegetables, fruits, beverages, and dairy products, in the respiratory, intestinal, and genital tracts of humans and animals, in sewage, and in plant materials [4]. They are normally found in nutrient-rich environments. They require fermentable carbohydrates, amino acids, fatty acids, salts, and vitamins for their growth [5] and compete actively and efficiently with other microbial species for the nutrients, resulting in a substantial enhancement in their viability. This results in the enhancement of the metabolic activities, causing the production of the desired type of metabolites possessing an inhibitory effect on the food-spoilage and pathogenic microorganisms, such as *Escherichia coli* and *Staphylococcus aureus*. Since homofermentative LAB can produce lactic acid from various types of fermentative carbohydrate sources present within meats, the homofermentative LAB is predominantly used in the mechanism of meat preservation. Bacteriocins produced by the group of LAB can be used as efficient preservative of meat and meat based products [3]. The present review deals with the use of LAB for the preservation and fortification of meat and meat products.

## 2. Bacteriocins and Their Classification

Bacteriocins are the group of ribosomally synthesized antimicrobial peptides. Over the years, various scientists provided the concepts of classification of LAB bacteriocins. LAB-produced bacteriocins are small, heat-stable, amphiphilic, and membrane-permeabilizing agents. These LAB bacteriocins can be majorly classified into three classes. The anionic cell wall of the bacteriocins producing bacteria possesses lipoteichoic and teichoic acids, which play an important role in the initial interaction of these anionic bacteriocins. Moreover, these bacteriocins have a higher range of antimicrobial activity at a lower pH as the properties, and the cell wall of the bacteriocin are pH-dependent. The three major classifications of the LAB bacteriocins are the lantibiotics, the non-lantibiotics, and the bacteriocins [6].

## 3. Genes Responsible for Bacteriocin Production

The performance of bacteriocin containing N-Terminal leader sequence is encoded by the structural bacteriocin gene, which helps prevent the activation of bacteriocin when present within the producer cell and helps recognize the transporter system (Figure 1).

The start codon codes for glycine that comprises 14–30 residues. The consensus elements in the double glycine leader comprise two glycine residues present at the C-terminus of the cleavage site, conserved hydrophobic and hydrophilic residues that remain separated by defined conserved residues. The double glycine leader comprises minimum of 14 amino acids. Among the consensus, residues being present within the leader, only the glycine at the −2 positions remains fully conserved. The mature bacteriocins are determined by a length of 30 residues but not more than 100 residues. It has been observed that colicin V comprises 80 residues that are being produced by *Escherichia coli*, which is formally grouped under class II bacteriocin [7]. Usually, bacteriocins comprise double peptides, both of which comprise double glycine leader and contain contiguous genes within the same operon. The peptides cannot be structurally differentiated from one another, and both are required for the optimal activity of the bacteriocin. The antimicrobial efficacy of Lactococcin MN and Lactococcin G is completely dependent upon the peptides being present at the leader sequence [8].

The bacteriocins under class II also share some common features like possessing high contents of amino acids like glycine. They are highly cationic, possessing a pI within the range of 8 to 11, and comprise the hydrophobic domain and amphiphilic region that provide its action upon the membranes [7]. The activity of Lacticin F is regulated by two functional genes, *tafA* and *lafX*, which enhance the ability of its inhibitory effect on other organisms.

## 4. Mechanisms of Action of Bacteriocin

There are several different mechanisms of bacteriocin action by which it exhibits its antimicrobial action to kill a pathogen or prevent food spoilage (Figure 2). The PMF (Proton Motive Force) Depletion mechanism, the Membrane Insertion & Pore Formation mechanism, and the Genetic control mechanism play an effective role in enhancing their activity or function.

### 4.1. PMF (Proton Motive Force) Depletion: Microorganism Bears an ATPase in Their Plasma Membrane

This bound ATPase plays a crucial role in aerobic and anaerobic energy transduction pathways. The role of the bacteriocins as observed in the research studies states that there are bacteriocins from the lactic acid bacteria, namely, Pediocin PA-1, Leuconocin S, and Lactacin F [9]. Therefore, these lactic acid bacteriocins play their antibacterial role by depletion of the PMF insensitive microorganisms. PMF is developed due to pumping out of electrons from the respiratory chains. As per research, the Pediocin PA-1 and Leuconocin S are responsible for the dissipation of the majority of *Listeria Monocytogenes*. In contrast, the Lactacin F plays a role in the 87% depletion the microorganism *Lactobacillus delbrueckii*. Though the Pediocin PA-1 and Leuconocin S are energy independent, Nicin is energy dependent. Thus, it has been the LAB bacteriocins share a similar process of depletion of PMF as depletion of PMF will inhibit several energy-demanding processes involved in bacterial membrane, including ATP synthesis; as a result of depletion, it will lead to the death of the harmful pathogen, and therefore, bacteriocin exhibits its antimicrobial or anti-bacterial properties through this mechanism [7].

### 4.2. Membrane Insertion and Pore Formation

*Listeria monocytogenes* and *Bacillus cereus* are a major group of pathogenic bacteria causing bacterial infections and meat spoilage. Therefore, antibiotics play a very important role in preventing these food spoilages. However, the demand for natural antibiotics due to the decrease in the chemical preservatives present in commercial antibiotics has led to the development of natural antibiotics or bacteriocins. Among such is Pentocin MQ1. The purification involved the absorption-desorption of the bacteriocin. This type of bacteriocin exhibited a wide spectrum of antimicrobial activity, high chemical and thermal activity, and a more stable pH. The mechanism involves cell membrane permeabilization or membrane insertion by pore formation.

LAB bacteriocins exhibited their anti-microbial properties mainly by pore formation and by inhibiting cell wall biosynthesis. For some cases, a high intensity of the fluorescence was observed within 30 min, but for some cases, it took a long time; however, the fact is that nisin adapts the pore formation mechanism to kill pathogens, and it is a more rapid killing mechanism. These bacteriocins usually have a wide range of activity, and they bear a stable pore; however, class II bacteriocins being small and heat-stable have a narrow range of activities. As mentioned, these bacteriocins cause membrane interaction and pore formation with the anionic lipid bilayer that is present abundantly in the membrane of the Gram-positive bacteria. Their activities are mainly enhanced by the presence of docking molecules and receptor that makes the pore stable. While the Lantibiotics produce wedge-like nodes, similarly, class II bacteriocins produce barrel stave pore with the help of carpet mechanism [8,10]. This mechanism, a single peptide molecule orients itself parallel to the surface of the cell membrane and brings about interference with the membrane bilayer without aggregation of the peptide molecule. If aggregation of peptide occurs in sufficient proportion, collapsing of the cell membrane takes place with the degradation of the membrane phospholipid [11].

There are certain steps that go along with pore formation. These are the proper steps that occur before a stable pore is established:**Membrane Interaction**—Gram-positive bacteria are known for their abundance of anionic lipid present in their membrane. Because of the cationic nature of the bacteriocin, the anionic lipid binding is enhanced. Nisin interacts to the anionic liposome of the Gram-positive bacteria. This interaction is strong because nisin binds slowly to other liposomes and the fragments of the nisin helps in identification of the region suitable for binding. Similarly, the class II bacteriocins initially follow the anionic phospholipid membrane interaction [12].**Membrane Insertion**—The insertion with the lipid monolayer has been established. As per the research study, the capability of the nisin to interact with the lipid layer present in the membrane is enhanced by the presence of strong antimicrobial activity. It has been stated that a low anti-microbial activity results in a decreased lipid interaction of the bacteriocin nisin and vice versa [12].**Cell Wall Passage and Receptors**—The class I bacteriocin Lantibiotics can form ion-conducting pores in the black lipid membranes of the Gram-positive bacteria. This results in the interaction with the peptidoglycan precursor lipid 2 in the membrane. The presence of the lipid 2 precursor enhances the ability of the bacteriocin to depolarize the electrical potential of the membrane. These data support the fact that lipid 2 acts as the docking molecule or the membrane nucleus, which controls all the activities. The Gram-positive cell wall allows the passage of the bacteriocins as enhanced by the Lipid 2. The polymers on the cell surface, lipoic acid and lipoteichoic acid, play a crucial role in the initial interaction of the bacterial cell wall and the bacteriocins. However, the receptor plays a role in this whole process and their implications have been explained in many research studies on specific narrow targets [13].

On the other hand, pore formation has been proposed, but particularly for the Lantibiotics-Nisin pore formation, it follows a series of specific distinct steps. These are the first arrangement of nisin that occurs parallelly with the membrane of the Gram-positive bacteria. With the insertion of the membrane of the Nisin domains, the cis acid and the trans negative can establish pore formation. The entire Nisin molecule moves across the membrane as the C-terminus inserts deeply into the membrane. Insertion of C-terminus causes intra-layer attachment with the phospholipids, resulting in the transmembrane movement of the nisin across the phospholipid. A PMF is required and is essential for creating the wedge-like model by nisin for pore formation. Creating a wedge-like model by nisin allows bending of the C-Terminal part, thereby assisting in membrane insertion. Inserting multiple Nisin molecules create a huge local dislocation and a large disruption in the lipid bilayer. As disruption is created in the membrane of the organism, it results in the formation of transient pores of lipid.

### 4.3. Bacteriocin Affecting DNA Synthesis

The insertion of the bacteriocin is generally followed by the inhibition of DNA synthesis in a Gram-positive bacterium. The inhibition of protein and RNA synthesis is known to be little affected. At very high concentrations of the bacteriocin megacin C, the destruction or inhibition of DNA, as well as protein and RNA synthesis, are quite rapid. However, prior treatment of the bacterium with chloramphenicol and streptomycin decreases the megacin action and limits its activity. Bacteriocins are protein peptides that bear high anti-bacterial activity that occurs as a chemical component within the cell components of the producing bacteria. It has been established that a single particle is strong enough to kill a strain of sensitive bacteria to the specific receptor sites present in the cell wall. The bacteriocin shuts down the DNA and RNA synthesis without affecting the respiration of the microorganism or the cellular mechanism [13]. Complete inhibition of DNA and RNA synthesis in sensitive cells, with a single particle of colicin. The antimicrobial action of bacteriocin megacin C was established and concluded that the killing activity of Megacin C is similar to that of a lethal agent. Inhibition of DNA was observed as the concentrations of megacin applied were higher. Similarly, the protein and nucleic acid synthesis were also inhibited when the concentrations were increased; however, the effect was indirect, which was consistent with the DNA dependent RNA synthesis.

## 5. Spoilage of Meat and Meat Products

Different types of protective and starter cultures are being used for a large number of byproducts used for safe-guarding the quality of meat from microbial degradation of the meat. LAB plays a vital role in transforming a large number of poultry and agricultural byproducts into useful substances that can be safe for human consumption. Various statistics showed that contaminated meat products were responsible for the cause of campylobacteriosis along with salmonellosis infections in the presence of Shiga toxin-producing *Escherichia coli* (STEC) and yersiniosis [14]. It has also been observed that various deaths occurred due to the infection caused by *Listeria monocytogenes,* and the fatality appeared very high [15]. *Listeria monocytogenes* are found profoundly within the meat and meat products and are associated with various types of foodborne diseases, attracting concerns [16]. Various studies are being conducted to counteract the enhanced condition of spoilage of meat and meat products and also preventing the use of chemical preservatives. Various physical processes, such as thermal processing, are performed for the purpose of meat preservation. Still, it has been observed that extensive heat results in the degradation of useful nutrients resulting in compromised quality of the products [17]. The meat spoilage occurs in ways that comprise souring, greening, and sliminess. The formation of biogenic amino acids takes place by the mechanism of decarboxylation of the amino acids during the manufacture and storage of sausages. These mainly occur in the presence of bacterial species such as *Pediococcus*, *Enterobacteriaceae*, *Streptococcus*, *Lactobacillus*, and *Bacillaceae*. It has been observed that poultry farms are the places that set the maximum outbreak of food poisoning and act as the best-suited culture media for bacteria [10].

*Enterobacteriaceae*, *Pseudomonas* spp., *Shewanella putrefaciens, Lactobacillus* spp., *Brchothrix thermosphacta*, and *Carnobacterium* spp. are the most predominant bacterial species that are responsible for the spoilage of pork and beef. The major changes that occur during the spoilage of meat comprise discoloration, off-flavors and production of gas, and alteration of pH. They mainly comprise *Lactobacillus* spp., *B. thermosphacta, Leuconostoc* spp., and *Carnobacterium* spp. The presence of the varied spectra of chemical compounds is used for the purpose of analyzing the spoilage of the meat and meat-associated products. The presence of diverse chemical within the spoiled meat and meat products are due to the microbe–microbe interactions, resulting in quality degradation [18].

The spoilage of meat occurs by decomposition and formation of various metabolites that occur during the growth of the microorganism. The microbiota existing within the meat brings about organoleptic changes. Thus, determining the spoilage of meat is very important, either by direct or indirect methods [19].

## 6. Development of Biofilm by LAB

Biofilms are considered the sessile microbial colonies that adhere to biotic and abiotic surfaces with the help of self-secreted polymeric substances [20] (Table 1). Lactobacilli are the group of Gram-positive cocci, rod, or cocco-bacilli found predominantly within food or feed products, having a high content of guanine-cytosine (GC) base-pairs and can perform carbohydrate metabolism [21]. Various types of LAB genera are associated with biofilm formation. The formation of biofilm by the probiotic bacterial species such as *Lactobacillus* appears to be beneficial, since it helps in promoting colonization and persistence on the mucosa layer of the host [22]. Various studies have been performed on the mechanism of biofilm development by LAB [23]. The EPS produced by these strains possesses the ability to inhibit the other biofilm developing pathogenic organisms [24].

## 7. Role of LAB in the Preservation of Meat and Meat Products

The use of LAB in the biopreservation of meat has provided a new arena in food preservation (Table 2). The studies were mainly performed in the use of LAB to eliminate various food-spoiling bacteria such as *E. coli* and *Salmonella* spp. *Lactobacillus salivarius* can inhibit *L. monocytogenes* and various types of *Salmonella* spp. thrive on the surface of the meat. Studies have also shown that *L. salivarius* does not degrade the quality of meat [28]. The biopreservation by LAB is controlled considerably in temperature [29]. It has also been observed that *L. sakei* helps control the growth of *Salmonella enterica serovar Choleraesuis* within fresh pork [30]. The combinatorial effect of LAB and *Pediococcus pentosaceus* helps in the marked reduction in the growth of *Staphylococcus* spp. within the raw meat [31]. It was observed that with the application of *P. acidilactici*, there was a reduction in the pathogenic count within the meat stored.

Still, the reduction in the pathogenic organism was found to be more when there was the combinatorial organism of *P. pentosaceus* and *P. acidilactici* was applied [34]. Higher temperatures, pH, and free fatty acids facilitate the protective activity exhibited by LAB [34]. The maintenance of pH below 5 by the LAB group is responsible for preventing the growth of pathogenic organisms in the meat. The lower amounts of thiobarbituric acid and free fatty acids within the cultures of LAB help in the lipolytic effect and oxidative damage of fat, thereby helping in the maintenance of freshness within beef. Bacteriocins produced from *Pediococcus acidilactici* when being applied to the meat of Turkey prevented the proliferation of the unwanted microbes such as *L. monocytogenes*, keeping it fresh for a longer period [1]. Studies have shown that a higher storage temperature facilitates the preservative action of the LAB in comparison to that of a lower storing temperature [38]. The planktonic and biofilm growth of *Lactobacillus* can act against *Candida parapsilosis*, a strain spoiling the meat products [39].

### Mechanism of Protection of Meat and Meat Products by LAB

LABs are considered safe microorganism having the ability to produce various types of inhibitory compounds like organic acids, hydrogen peroxides, N-diacetyl, carbon dioxide, and bacteriocins. It can bring about inhibition of various types of harmful organisms by a competitive exclusion mechanism for various binding sites and nutrients. It has the ability of various enzymatic functions to aid in better nutrient utilization and the stimulation of the immunity in animals [40].

The meat-associated LAB can produce hydrogen peroxide that acts as a protective mechanism for preventing the damage by other pathogenic organisms [41]. Hydrogen peroxide produced by LAB acts as bacteriostatic for Gram-positive bacteria and bacteriocidal for Gram-negative bacteria [9]. Certain groups of LAB also possess the ability to produce biogenic amines by decarboxylation of the amino acids [42]. The presence of biogenic amines act as index for understanding the meet quality and stability [43].

Pediocin such as bacteriocins and pediocins exhibit antimicrobial potential and act as a potential antilisteria agent for biopreservation of meat and meat-associated products [44]. Various studies have shown that the application of bacteriocin-associated strains, and bacteriocin helps enhance the shelf-life of meat and meat products [45]. The use of bacteriocins in packaging also help in improving the safety of meat and meat products [2]. Sakacin G is a type of bacteriocin produced by *Lactobacillus curvatus* that plays an important role in preserving food and enhancing the shelf-life and safety of the meat and meat-associated products. It is being used as artificial and natural products that help preserve a wide range of meat. The major advantage of the bacteriocin is that it is used as an extra wrapping for supporting various types of antimicrobial efficacy [46].

The bacteriocins produced from the LAB are used as an important ingredient of food and act as an important source of food preservative within that act as ready-to-eat products. They have shown their efficacy as potent antimicrobial agents and prevent the growth of meat spoiling organisms [47] (Figure 3).

## 8. Fortification of Meat Products by LAB

Various types of probiotic bacteria help produce various nutraceuticals and micronutrients that help in the in situ mechanism of fortification of meat and meat-associated products. This results in enhancing the nutritional quality of the product [48]. The use of probiotics has various types of advantages that comprise improvement in the symptoms associated with lactose intolerance, enhancement of the immune responses, improvement in the digestion and intestinal transit, reduction of diarrhea, reduction in the chances of the development of colon cancer, and reduction in the level of cholesterol [49]. This has resulted in the use of probiotic organisms in meat preservation and safety for human consumption. The probiotic has shown its utility in raw fermented products such as salami. The probiotic properties associated with the LAB are used within the fermented meat products. These cultures provide technological and sensory characteristics and have a beneficial effect on the health of the consumers. It has been observed that salami comprise three important intestinal LAB groups like *Lactobacillus casei* and Bifidobacterium spp. Greater performance of fermentation of meat can be achieved by the group of *Lactobacillus acidophilus* bacteria comprising *L. crispatus*, *L. amylovorus*, *L. johnsonii*, *L. gasseri*, and *L. acidophilus.* The organisms were able to exhibit the greatest fermentation performance by preventing bile and gastric juices that usually have a detrimental effect in the intestine. This also prevents the growth of *Staphylococcus aureus*, thereby preventing the production of enterotoxin and helping produce higher quality meat and meat products [50]. *Lactobacillus sakei* produces lactocin S that can protect ham associated products, thereby helping to establish the biopreservative potential [51].

## 9. Conclusions and Future Perspective

The use of LAB in fermentation has a long historical background. However, until now, there was a dearth of studies on the process of utilization of LAB as biopreservation agents. Various studies have proclaimed that LAB showed its efficiency in preserving meat and meat-associated products. Only few purified bacteriocins have been officially approved by the FDA as a meat preservative. The antimicrobial metabolites produced by LAB show a very high potential in protecting the meat and associated products from microbial damage, thereby meeting the consumer demand for food safety and security. They also help enhance the shelf-life, prevent the growth of the pathogenic organisms, and also help in providing the sensory characteristics of meat products. They also act as a potential alternative to fight against various resistant organisms. The metabolites produced by the LAB act as a suitable alternative and possess the ability to solve various economic losses that the industry suffers from due to spoilage of meat and meat products. The group of LAB produces antimicrobial peptides those act as potential alternatives to the commonly used additives that help in standardizing the quality of meat in accordance to the need of the consumers. They also help in the enhancement of the shelf-life of the food and food products in comparison to the normal additives those are used for the preservation. Additionally, they help in protecting the meat products from various types of food borne pathogens and also help in providing sensorial characteristics to various meat products. Large economic loses can be solved by the use of the biopreservation technique. It has been observed that the use of the LAB in the real meat results in lowering the antibacterial activity due to the complexity of the food products. Some problems also arise when upscaling the same at the industrial level. The major challenge at recent point of time is the regulatory framework which can tangle the use of novel bacteriocins as food additives. Their performance is dependent of the temperature and time of storage, range of pH, and its interactions with various components of the microbiota associated with the food. Thus, more research needs to be performed in the mechanism of improving their use either in the form of single dosage or in combinatorial effect. The use of biopreservatives along with various other preservation techniques help in providing better results in context to the technique of preservation of meat and meat associated products. Until now, there was a dearth of research in the context to the combinatorial approach of the use of LAB or bacteriocins in preservation of meat and its products. The use of omics in the identification of microorganisms from the environment is associated with food processing and the microbiome of the food. These recent technologies would help in facilitating the various novel biopreservation strategies and also assessing the efficacy of their action. Thus, these techniques would definitely emphasize food safety techniques.

## Figures and Tables

**Figure 1 foods-11-02792-f001:**
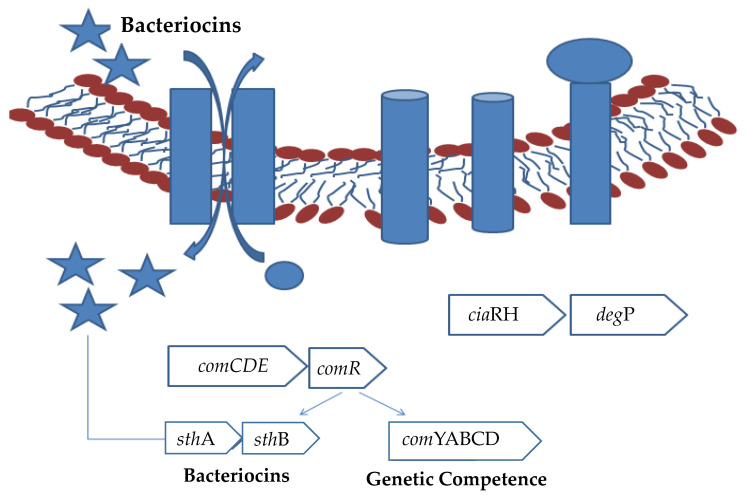
Genetic regulation in the production of bacteriocin production by LAB. Upregulation of the genes results in the production of the bacteriocins. The star symbols indicate bacteriocins. Circles indicate the proteins being produced within the cell.

**Figure 2 foods-11-02792-f002:**
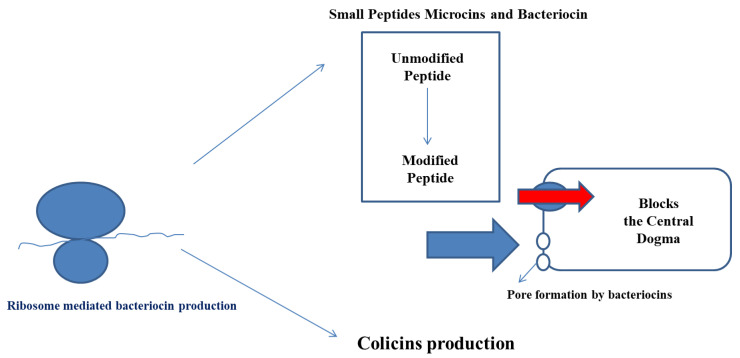
Mode of action of the bacteriocins in preventing the growth of other microbial species.

**Figure 3 foods-11-02792-f003:**
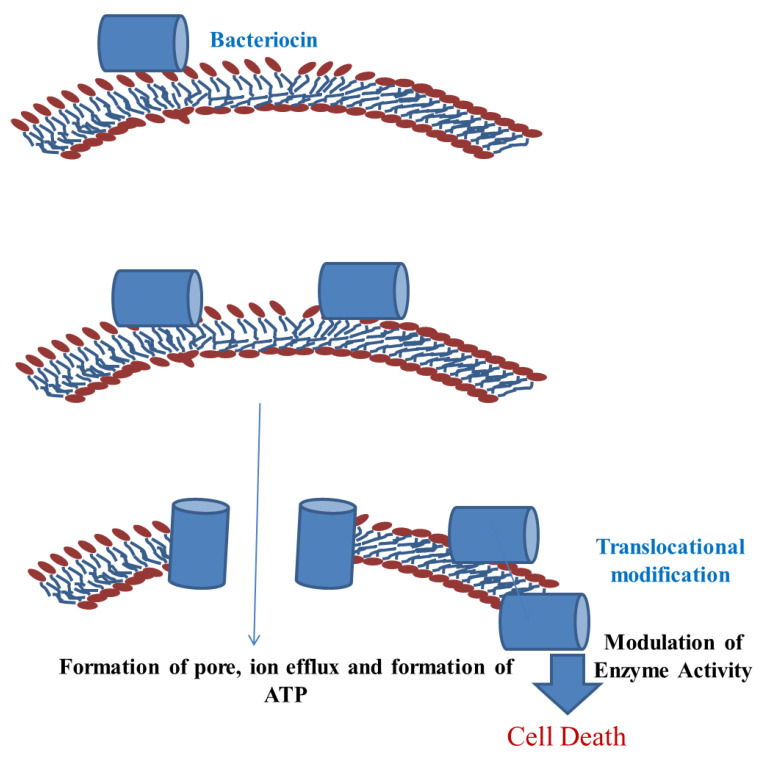
Mechanism of inhibiting the pathogenic cells responsible for meat degradation.

**Table 1 foods-11-02792-t001:** Biofilm development by LAB.

Name of LAB	Function of the Biofilm	Reference
*Lactobacillus rhamnosus*	The development of the biofilm helps in the inhibition of the biofilm formed by *Salmonella* sp. H9812 and *Escherichia coli*	[25]
	*welE* gene helps in the production of the exoplysaccharide (EPS) there by helping in the adherence with various surfaces	[26]
*Lactiplantibacillus plantarum*	Helps in the eradication of organisms like *Salmonella enterica serovar Enteritidis*, *Staphyloccocus aureus*, and *Listeria monocytogenes*	[27]
*Lactobacillus reuteri*	Inhibits the growth of *Escherichia coli,*	[27]
*Lactobacillus fermentum*	Helps in inhibiting the growth of *S. Enteritidis* and *Escherichia coli*	[27]

**Table 2 foods-11-02792-t002:** LAB-associated protection of various types of packaged meat and meat products.

Preserved Meat	Type of LAB Used	Type of Targeted Organism	Observation	Reference
Raw beef packed in vacuum packs	*L. curvatus*-mediated production of lactocin	*B. thermosphacta*	Effective viability reduction of *B. thermosphacta* was observed	[32]
Pork-ham ready to eat packs	*P. pentosaceus*-mediated production of bacteriocin-like substance and nisin	*L. seeligeri*	It helps in bringing about log 1.7 times inhibition of *L. seeligeri*	[12]
Meat balls made up of beef	*L. plantarum*-mediated production of bacteriocin	*E. coli* and *Salmonella enterica* serovar *Typhimurium*	It helps in the significant reduction of the pathogenic organism	[31]
Beef slices	*C. maltaromaticum*-mediated production of bateriocin	*S. Typhimurium* and *E. coli*	It brings about marked reduction of the targeted organism	[33]
Fresh samples of beef	*P. acidilactici* and *P. pentosaceus*	*S. Typhimurium* and *L. monocytogenes*	Brings about two-fold reduction in the growth of the pathogenic organisms those are associated with the degradation of meat	[34]
Sausages of meat	*P. acidilactici*-associated production of bacteriocin	*L. monocytogenes*	Three-fold reduction in the targeted organism	[16]
Sucuk sausages	*L. plantarum*-mediated production of bacteriocin	*L. monocytogenes*	Brings about marked reduction in the growth of the microorganims	[35]
Emulsion of goat meat	*Murraya koenigii* andb *P. pentosaceus*-mediated pediocin production	*L. innocua*	Brings about 2- to 3-fold reduction in the targeted organism	[36]
Natural casings of sheep	Bacteriocins produced by LAB	*Clostridium sporogene*	Brings about marked reduction in the *Clostridium* sp.	[37]

## Data Availability

No new data were created or analyzed in this study. Data sharing is not applicable to this article.

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
