# Peer review of "Lactic Acid Bacteria (LAB): Autochthonous and Probiotic Microbes for Meat Preservation and Fortification"

_foods, 2022, doi:10.3390/foods11182792_

Round 1

Reviewer 1 Report

This narrative review is sufficiently well structured, although personally I prefer a structure similar to an experimental work, which clearly indicates a process intended to answer a specific research question, as in systematic reviews; I would recommend that Authors set up their review this way.

Author Response

Reviewer’s comment

Author’s response

This narrative review is sufficiently well structured, although personally I prefer a structure similar to an experimental work, which clearly indicates a process intended to answer a specific research question, as in systematic reviews; I would recommend that Authors set up their review this way.

Thank you for reviewing and positive evaluation of our MS.

We have incorporated the changes

Reviewer 2 Report

The manuscript is a review that deals on the use of LAB as biopreservatives for enhancing the shelf-life of meat and meat products and also helps in its fortification. The manuscript has some merit although I believe that the novelty should be more clearly stated in the introduction. For example, the Introduction section at no time is demonstrated "how the use of the LAB can significantly is predominantly used in the mechanism of meat preservation?" 

The item "Bacteriocins and their classification" does not need to be subdivided. It becomes necessary for the authors to demonstrate results with these different identifications, comparing and demonstrating advantages and disadvantages of each one. The form presented looks more like a teaching material than a review. Furthermore, this idem seems more like a copy of the work of Reference [6].

The figures illustrate the Review well, however they need a better discussion. They are just presented without contributing more intensively to the work.

The Results presented in Table 1 of item 6 "Development of biofilm by LAB" need to be improved. The authors present only the "Function of the Biofilm". it's little. And the applicability? What kind of food was it used on? What are the main findings?

Table 2? This is not a table. Table do not have side lines.

 The presentation of results needs to be improved. The interpretation and discussion of the results are somewhat repetitive and speculative and should be revised to be more succinct and focused on what was observed. However, there are several major concerns that need to be addressed to consider this manuscript for publication.

Author Response

Reviewer’s comment

Author’s response

The manuscript is a review that deals on the use of LAB as biopreservatives for enhancing the shelf-life of meat and meat products and also helps in its fortification. The manuscript has some merit although I believe that the novelty should be more clearly stated in the introduction. For example, the Introduction section at no time is demonstrated "how the use of the LAB can significantly is predominantly used in the mechanism of meat preservation?"

Thank you for reviewing and positive evaluation of our MS.

We have incorporated the changes

The item "Bacteriocins and their classification" does not need to be subdivided. It becomes necessary for the authors to demonstrate results with these different identifications, comparing and demonstrating advantages and disadvantages of each one. The form presented looks more like a teaching material than a review. Furthermore, this idem seems more like a copy of the work of Reference [6].

We have removed the same, and corrected the text according to the reviewer´s suggestion.

The figures illustrate the Review well, however they need a better discussion. They are just presented without contributing more intensively to the work.

Following the reviewer suggestion, we incorporated the changes.

The Results presented in Table 1 of item 6 "Development of biofilm by LAB" need to be improved. The authors present only the "Function of the Biofilm". it's little. And the applicability? What kind of food was it used on? What are the main findings?

Incorporated the changes.

Table 2? This is not a table. Table do not have side lines.

Table was corrected.

The presentation of results needs to be improved. The interpretation and discussion of the results are somewhat repetitive and speculative and should be revised to be more succinct and focused on what was observed. However, there are several major concerns that need to be addressed to consider this manuscript for publication.

The text was corrected and improved, following the reviewer indication.

Reviewer 3 Report

Line 54, might also add other pathogenic strain such as E. coli O157:H7, S. aureus.

Iine 65 please follow the author instructions regarding the cite reference.

Line 100, please mention the conserved sequence in bracket.

Line 113, what is the average size of most bacteriocins?

Figure 1 is unclear, please improve the quality of the figure in terms of the symbols such as the stars and the circle, what is the meaning of the stars and circle? Please describe it in the Figure 1.

Line 122, what do you mean by “the leader glycine”? did you mean that the start codon is glycine?.

Line 165, change Cereus to cereus.

Line 185 what is carpet mechanism? Please describe it

Line 240, what is the concentration of bacteriocin that is considered high as antibacterial?

Line 264 what type of bacteria in poultry?

Line 273, what do you mean “purification” during microbe-microbe interaction?

Line 278, different font size from line 279-287.

It would be nice if the authors can provide a list of pure bacteriocin that have been successfully applied in food system, maybe the authors can provide the name of bacteriocin, type of bacteriocin, concentration, application in the food system.

Author Response

Reviewer’s comment

Author’s response

Line 54, might also add other pathogenic strain such as E. coli O157:H7, S. aureus.

We have added in the manuscript

Line 65 please follow the author instructions regarding the cite reference.

We have modified the reference as per author instruction

Line 100, please mention the conserved sequence in bracket.

We  have removed the portion as per suggestion of Reviewer

Line 113, what is the average size of most bacteriocins?

We  have removed the portion as per suggestion of Reviewer

Figure 1 is unclear, please improve the quality of the figure in terms of the symbols such as the stars and the circle, what is the meaning of the stars and circle? Please describe it in the Figure 1.

We have incorporated the changes

Line 122, what do you mean by “the leader glycine”? did you mean that the start codon is glycine?

We have incorporated the changes

Line 165, change Cereus to cereus.

We have incorporated the changes

Line 185 what is carpet mechanism? Please describe it

We have incorporated the changes

Line 240, what is the concentration of bacteriocin that is considered high as antibacterial?

The concentration varies from one type of bacteriocin to other

Line 264 what type of bacteria in poultry?

Enterobacteriaceae, Pseudomonas spp., Shewanella putrefaciens, Lactobacillus spp., Brchothrix thermosphacta and Carnobacterium spp.

Line 273, what do you mean “purification” during microbe-microbe interaction?

We have incorporated the changes

Line 278, different font size from line 279-287.

Incorporated the changes

It would be nice if the authors can provide a list of pure bacteriocin that have been successfully applied in food system, maybe the authors can provide the name of bacteriocin, type of bacteriocin, concentration, application in the food system

Thank you for the suggestions. There are still dearth of work done in the field of meat preservation. We mentioned very few in accordance to the work that has been performed.

Round 2

Reviewer 2 Report

The authors did a good job and responded to review requests.

Author Response

Thank you for your comment. The manuscript was checked according to editor's comment.

Reviewer 3 Report

the authors have improved the manuscript, no further revision from my side

Author Response

Thank you for your comment. The manuscript was checked according to the editor's comment.